# A Correlation between 3′-UTR of *OXA1* Gene and Yeast Mitochondrial Translation

**DOI:** 10.3390/jof9040445

**Published:** 2023-04-05

**Authors:** Maryam Hajikarimlou, Mohsen Hooshyar, Noor Sunba, Nazila Nazemof, Mohamed Taha Moutaoufik, Sadhena Phanse, Kamaledin B. Said, Mohan Babu, Martin Holcik, Bahram Samanfar, Myron Smith, Ashkan Golshani

**Affiliations:** 1Department of Biology and Ottawa Institute of Systems Biology, Carleton University, Ottawa, ON K1S 5B6, Canada; 2Department of Biochemistry, Research and Innovation Centre, University of Regina, Regina, SK S4S 0A2, Canada; 3Department of Pathology and Microbiology, College of Medicine, University of Hail, Hail P.O. Box 2240, Saudi Arabia; 4Department of Health Sciences, Carleton University, Ottawa, ON K1S 5B6, Canada; 5Agriculture and Agri-Food Canada, Ottawa Research and Development Centre (ORDC), Ottawa, ON K2H 8S2, Canada

**Keywords:** gene expression, messenger RNA, translation, yeast, mitochondria, untranslated regions

## Abstract

Mitochondria possess their own DNA (mtDNA) and are capable of carrying out their transcription and translation. Although protein synthesis can take place in mitochondria, the majority of the proteins in mitochondria have nuclear origin. 3′ and 5′ untranslated regions of mRNAs (3′-UTR and 5′-UTR, respectively) are thought to play key roles in directing and regulating the activity of mitochondria mRNAs. Here we investigate the association between the presence of 3′-UTR from *OXA1* gene on a prokaryotic reporter mRNA and mitochondrial translation in yeast. *OXA1* is a nuclear gene that codes for mitochondrial inner membrane insertion protein and its 3′-UTR is shown to direct its mRNA toward mitochondria. It is not clear, however, if this mRNA may also be translated by mitochondria. In the current study, using a *β*-*galactosidase* reporter gene, we provide genetic evidence for a correlation between the presence of 3′-UTR of *OXA1* on an mRNA and mitochondrial translation in yeast.

## 1. Introduction

Diverse functions of mitochondria can vary from cell to cell. In addition to its function in signaling pathways and metabolism, the most predominant role of mitochondria is to produce energy for cell survival in the form of adenosine triphosphate (ATP). Mitochondria’s activities are facilitated by mitochondrial proteins encoded by either nuclear DNA or maternally-inherited mitochondrial DNA (mtDNA) [1,2]. Like nuclear DNA, mtDNA goes through DNA replication, transcription, and translation [3]. The machineries involved in these processes, however, resemble those of prokaryotic organisms, reiterating the evolutionary origin of mitochondria [4]. DNA damage, heat shock, UV radiation, and other stimuli, including chemotherapeutic drugs, can affect mitochondria-driven apoptosis (MDA) [5].

In a number of neurological disorders, including Parkinson’s and Alzheimer’s diseases, mutations in the respiratory chain and oxidative phosphorylation system (OXPHOS), encoded by mtDNA, can lead to premature cell death [6]. Patients with certain mitochondrial disorders are also shown to have impaired immune systems and a higher probability of infections [7]. Other mitochondrial-associated disorders include diabetes and a number of cancers [8]. Targeting mitochondrial biosynthesis and limiting mitochondrial reactive oxygen species (ROS) production is thought to be a promising cancer therapeutic strategy [7]. In this context, targeting different agents, including proteins, DNA, mRNA, nano-agents and antioxidants to mitochondria. has been proposed for therapeutic purposes [9,10].

In humans, *Saccharomyces cerevisiae* mtDNA carries a number of genes involved in gene expression and the respiratory chain, in addition to several ribosomal proteins and tRNAs [11]. However, a majority of the proteins that function within this organelle are of nuclear origin and are imported to mitochondria [1,2]. Certain nuclear RNAs, including 5S ribosomal RNA and microRNAs, are also reported to enter mitochondria [12]. The detailed mechanism of these imports is not clear and requires further investigations [13]. In a yeast genome-wide analysis, 466 nuclear genes were identified that, when deleted, caused impaired mitochondria respiration [14]. Human homologs of many of these genes were linked to human mitochondria disorders, indicating that yeast may serve as a suitable model organism to study mitochondria associated human diseases [13].

It is generally accepted that localization of certain nuclear mRNAs to the vicinity of mitochondria leads to the localized translation of these nuclear genes near to the mitochondria. Localized translation is generally attributed to cytoplasmic translation. It is thought to be followed by the import of the newly translated proteins into mitochondria [1,2]. An example of these mRNAs is that of *OXA1* gene, which codes for mitochondria inner membrane insertion protein. The 3′-UTR of the *OXA1* gene was shown to direct the mRNA to the vicinity of mitochondria [15]. It is suggested that this 3′-UTR serves as a binding site for certain RNA-binding proteins that, with the help of motor proteins, transport mRNAs towards the mitochondria [1]. Then, *OXA1* mRNA is translated near the outer surface of the mitochondria. This is followed by the import of the newly synthesized Oxa1p into the mitochondria [16]. In this way, the presence of Oxa1p in mitochondria is directly influenced by the 3′-UTR of its mRNA. However, from the current data, it is not clear if *OXA1* mRNA molecules may also be translated by the mitochondrial machinery. It is also unclear if *OXA1* mRNA may also enter the mitochondria. A main reason for this lack of knowledge stems from technical limitations associated with purification experiments. An inherent limitation of purification procedures is that they all suffer from different degrees of co-purifying contaminants. Consequently, deriving precise conclusions from purification experiments alone is not an easy task. There is often a need for designing unique approaches to answer specific questions. *OXA1* mRNA is known to be abundant and efficiently translated in the cytoplasm of yeast. With currently available methodologies, it is challenging to differentiate whether a small portion of *OXA1* mRNA may also be translated by the mitochondria.

In the current study, we investigate the 3′-UTR of *OXA1* mRNA for its possible correlation with mitochondrial translation. We designed an approach that uses the foundations that govern mitochondrial translation as the basis to evaluate the translation of a prokaryotic reporter *β*-*galactosidase* mRNA via mitochondria. An important feature of our approach is that the construct is designed to hinder the cytoplasmic translation of its mRNA. Our observations provide genetic evidence for a correlation between 3′-UTR of *OXA1* mRNA and the translation of *β*-*galactosidase* mRNA by the mitochondrial translation machinery. Based on universal codon exceptions in mitochondria, mass spectrometry data also supports the synthesis of *β*-*galactosidase* mRNA by yeast mitochondria.

## 2. Materials and Methods

### 2.1. Strains, Plasmids and Media

The baker’s yeast, *S. cerevisiae* strain BY4741, was used in this study. The expression vector p416GALL [17] was utilized for expression analysis. This vector contains a galactokinase promoter (GALL-pro) upstream of a *lacZ* reporter gene that was cloned as an Xbal/BamHI cassette. The plasmid also contains Uracil (URA3) and Ampicillin resistance (Amp) genes for selective growth of yeast and *E. coli,* respectively, in selective media [18]. DH5α strain of *E. coli* was used to propagate plasmids. Standard rich (YPD) and synthetic complete (SC) media were used as growth media for yeast, unless specified otherwise. LB (Lysogeny broth) was used to grow *E. coli.* Antibiotics were used in the following concentrations: ampicillin 100 μg/mL cycloheximide 45 ng/mL and chloramphenicol 1.4 mg/mL.

### 2.2. DNA Manipulations

3′-UTR of OXA1 was amplified by PCR from genomic template using the following primers: 3′-YOXAlUTR-F (5′ CGCGGATCCATTAATAACAAAAAATGAATAAAGGC 3′) and 3′-YOXAlUTR-R (5′ CGCGGATCCTCCAAATGATTATTTCAAGCAATAAA 3′). The resulting PCR product was digested by BamHI and ligated into the unique BamHI site of p416GALL. Sequencing was used for confirmation. Designed 5′-UTRs were cloned by inserting synthesized double stranded DNA into the unique XbaI site of p416GALL plasmid. The following pairs of primers were synthesized and hybridized in ligation buffer. Primers 5COXa (5′ AATAGTATTAACATATTATAAATAGACAAAAGAGTCTAAAGGTTAAGATTTATTAAAATGC 3′) and 5COXb (5′ CTAGGCATTTTAATAAATCTTAACCTTTAGACTCTTTTGTCTATTTATAATATGTTAATACTATTACGCCATGGTCAGCTTACGCCCGCCTGTTTGGCGGGCGTAAGCTGG 3′) were used for MT-C1 construct and primers 5COXDa (5′ CTAGCCAGCTTACGCCCGCCAAA CAGGCGGGCGTAAGCTGACCATGGCGTAATAGTATTAACATATAGATCTTAGACAAAAGAGTCTAAAGGTTAAGATTTATTAAAATGC 3′) and 5COXDb (5′ CTAGGCATTTTAATAAATCTTAACCTTTAGACTCTTTTGTCTAAGATCTATATGTTAATACTATTACGCCATGGTCAGCTTACGCCCGCCTGTTTGGCGGGCGTAAGCTGG 3′) were used for the MT-C1X construct. Sequencing was used for confirmation. Plasmids were extracted from transformed cells via QIAprep Miniprep kit, using the manufacturer’s specifications.

### 2.3. Mitochondria Isolation

Yeast mitochondria were isolated by Sigma-Aldrich yeast mitochondrial isolation kit according to the manufacturer’s specification. Isolated mitochondria were treated by RNAse and further purified using a Free Flow Electrophoresis approach, as described in [19]. Mitochondrial fractions were pooled and concentrated by centrifugation.

We also used Yeast Mitochondrial Isolation Kit (ab178779) to extract mitochondrial protein and prepare samples for Mass spectrophotometry analysis.

### 2.4. qRT-PCR

cDNA synthesis of mRNA samples was processed according to the Bio-Rad RT-PCR kit. Quantitative PCR was performed using the iQSYBR Green master-mix kit (Biorad, Hercules, CA, USA) according to manufacturer’s instructions. qPCR amplification and detection were performed on the RT-PCR cycler (ROTOR GENE RG-3000 from Corbett research). *PGK1* housekeeping gene was used as a control. Data were analyzed using Rotor-Gene Real-Time Analysis Software 6.0.14.

### 2.5. β-Galactosidase Liquid Assay

Quantitative *β*-*galactosidase* assay was performed using the ONPG (O-nitrophenyl-a-D-galacto-pyranoside) method, as described in [17,20]. Cells were induced by exposure to 2% galactose for 6 h. As needed, cycloheximide or chloramphenicol was added to the induction media. When required, yeast mitochondria was isolated using BD Free Flow Electrophoresis (FFE) system, according to the manufacturer’s specifications [19].

### 2.6. Mass Spectrometry Analysis

Yeast strains carrying each of the constructs were grown overnight and the cells were induced by exposure to 2% galactose for 6 h the next day. The mitochondrial protein was extracted via Yeast Mitochondrial Isolation Kit (ab178779) from each sample and were dissolved into 6 M Urea, 2 M Thiourea and 50 mM Tris-HCl buffer (pH 7.5) with protease inhibitor cocktail I, Animal-Free (Millipore, Burlington, MA, USA), and heated at 60 °C for 1 h. Samples were diluted 4 times with 50 mM Tris-HCl buffer (pH 7.5) with 1 mM DTT. 20 μg/mL Trypsin (Promega, Madison, WI, USA) was added to each sample and the sample solution was incubated at 37 °C for 30 min. 2-chloroacetamide (0.5 mM, final concentration) was then added and the samples were incubated at 37 °C, along with shaking for 16 h. 2 μL of TFA was added to the samples to stop Trypsin digestion. Digested samples were then desalted with Top-Tip C-18 cartridge (Glygen). Top-Tip was wetted with wetting solution (60% acetonitrile, 0.1% formic acid in water) and then equilibrated by equilibration solution (2% acetonitrile and 0.1% formic acid in water). Peptide samples were loaded on the column and centrifuged at 500× *g* for 5 min. After washing by water with 0.1% formic acid solution, peptide samples were eluted by 100 μL of elution solution (60% acetonitrile, 0.1% formic acid in water). Eluted peptides were air-dried and dissolved in 20 μL of water with 0.1% formic acid.

All samples were analyzed by nLC coupled to the Orbitrap Elite mass spectrometer (Thermo Fisher Scientific, Waltham, MA, USA). Chromatographic separation of peptides was performed on a Proxeon EASY nLC 1000 System (Thermo Fisher Scientific) equipped with an in-house C18 column, 15 cm × 148 μm ID (Polymicro), 3 μm PRP-C18 Resin (Hamilton), employing a water/acetonitrile/0.1% formic acid gradient, and 5 μL of the samples were loaded onto the column for 210 min at a flow rate of 0.30 μL/min. Peptides were separated using 1% acetonitrile, increasing to 3% acetonitrile in the first 2 min, and then using a linear gradient from 3 to 24% of acetonitrile for 170 min, followed by a gradient from 24 to 100% of acetonitrile for 28 min and a wash at 10 min, at 100% of acetonitrile. Eluted peptides were directly sprayed into mass spectrometer using positive electrospray ionization (ESI) at an ion source temperature of 250 °C and an ion-spray voltage of 2.1 kV. Full-scan MS spectra (*m*/*z* 350–2000) were acquired in the Orbitrap elite at 60,000 (*m*/*z* 400) resolution. The automatic gain control settings were 1e6 for full FTMS scans and 5e4 for MS/MS scans. Fragmentation was performed with collision-induced dissociation (CID) in the linear ion trap when ions intensity was >1500 counts. The 15 most intense ions were isolated for ion trap CID with charge states ≥2 and sequentially isolated for fragmentation using the normalized collision energy set at 35%, activation Q at 0.250 and an activation time of 10 ms. Ions selected for MS/MS were dynamically excluded for 30 s. Calibration was performed externally with Pierce LTQ Velos ESI Positive Ion Calibration Solution (Thermo Fisher Scientific, catalog number 88322). The Orbitrap Elite mass spectrometer was operated with Thermo XCalibur software. All RAW files were converted to mzXML using ReAdW-4.3.1.

Thermo .raw files were converted to .mzXML using ReAdW-4.3.1 and searched against a trypsin digested database of canonical MT-C1 and MT-C2 target + decoy protein sequences using the Crux (v4.1) Tide search algorithm with default parameters. Two missed cleavages were allowed and carbamido-methylation of cysteine and oxidation of methionine were set as fixed and variable modifications. The precursor ion mass tolerance was set to 10 ppm, and the fragment ion mass tolerance was set to 0.2 Da.

## 3. Results and Discussion

mRNA targeting is generally regarded as a component of localized translation. The 3′-UTR of *OXA1* mRNA is believed to mediate the targeting of this mRNA to the vicinity of mitochondria where the mRNA is translated and subsequently imported [21,22]. *OXA1* codes for a mitochondria inner membrane protein thought to play a role in membrane binding of ribosome and the insertion of newly synthesized polypeptides into mitochondrial inner membrane [4]. There are also examples of numerous RNA molecules that are imported into mitochondria in different organisms. Various mitochondrial tRNAs are encoded by the nuclear genome and subsequently targeted into mitochondria [13]. More recently, a 20 nucleotide sequence from the *H1* RNA, the RNA component of RNase P enzyme, was shown to mediate the import of fusion tRNA and mRNA molecules into mitochondria and rescuing phenotypic defects in two different human cell lines caused by defective tRNAs [9]. *OXA1* 3′-UTR is shown to target mRNAs to the vicinity of yeast mitochondria. Here, we asked if this 3′-UTR might also mediate the mitochondrial translation. For this purpose, we designed a specific approach that utilized the foundations that govern mitochondrial translation and architected a distinct prokaryotic reporter *β-galactosidase* construct. This construct, termed MT-C1 for Mitochondrial Translation Construct 1, is designed to hinder cytoplasmic translation of its mRNA and also promote its translation via mitochondrial machinery (Figure 1). At its 3′-end, MT-C1 carries *OXA1* 3′-UTR. Its 5′-UTR is designed to impede cytoplasm translation by including the following elements: (1) a strong inhibitory secondary structure with estimated ΔG value of −28 kcal mol^−1^ previously shown to prevent the advancement of 40S ribosomes along mRNA and thereby inhibiting translation [23]. The purpose of this structure is to prevent the scanning of the cytoplasm ribosomes from 5′-CAP towards the start codon; (2) an early initiation start codon next to a consensus Kozak sequence. The presence of a strong Kozak sequence will promote an out-of-frame cytoplasmic translation initiation for those ribosomes that may escape the inhibitory structure above. These ribosomes will start translation from the out of frame AUG codon; (3) consecutive premature stop codons downstream of the early start codon mentioned above. These stop codons can terminate unwanted premature cytoplasmic translations. Mitochondrial translation is reported to require a consensus translation initiation sequence 5′-UAUAAAUA-3′. To promote mitochondrial translation, 5′-UTR from *COX2* mRNA, which contains the mitochondria consensus translation initiator sequence, was incorporated upstream of the AUG start codon of the reporter mRNA. *COX2* codes for cytochrome C oxidase subunit II, the terminal member of mt inner membrane electron transport chain, and is translated in mitochondria [24]. It is one of the major mitochondrial mRNAs and has a well-characterized mitochondrial 5′-UTR and the shortest (54 nucleotides) reported [24,25]. To further enhance mitochondrial translation, a prokaryotic *β-galactosidase* reporter gene was used as the reporter. A second construct termed MT-C2 was designed as a control. It contains all the features of MT-C1 with the exception that it lacks the 3′-UTR of *OXA1* at its 3′-end, thought to direct the mRNA towards the mitochondrial vicinity.

### 3.1. OXA1 Gene 3′-UTR Directs β-Galactosidase mRNA to the Vicinity of Mitochondria

To study if the 3′-UTR of *OXA1* mRNA directs a reporter *β-galactosidase* mRNA to the vicinity of mitochondria, the constructs MT-C1 and MT-C2 were used to transform yeast cells. Mitochondria were isolated from the transformed cells. The *β-galactosidase* mRNA content, as well as the *β-galactosidase* activities of the purified mitochondrial samples, were measured (Figure 2). Represented in Figure 2a, mRNA analysis using RT-qPCR indicated that *β-galactosidase* mRNA content for those mRNAs lacking *OXA1* 3′-UTRs (MT-C2) is approximately reduced by 90% in comparison to those carrying *OXA1* 3′-UTR (MT-C1). Similarly, *β-galactosidase* activity measurement showed an approximately 85% reduction for mRNAs lacking *OXA1* 3′-UTR. These results are in accord with the observation by others that *OXA1* 3′-UTR seems to direct its own mRNA toward the vicinity of the mitochondria and that the mRNA may be translated near to the mitochondria. This provides further evidence for a general localization property of this sequence toward mitochondria.

### 3.2. Observed β-Galactosidase Activity Contains Signatures of Mitochondrial Translation

To further study the correlation between the translation of mRNAs carrying *OXA1* 3′-UTR and mitochondrial translation, we designed a control construct termed MT-C1X. This new construct is identical to the MT-C1 construct, with the exception that its 8 nt *COX2* translation initiation signal that promoted mitochondrial translation within 5′-UTR of MT-C1 was deleted. In designing the MT-C1 construct, we have gone to some lengths to ensure limited cytoplasmic translation from this construct. However, it is still possible that the observed translation above might stem from cytoplasmic translation. Deletion of the *COX2* mitochondrial translation signal from MT-C1 5′-UTR can aid in distinguishing cytoplasmic translation from that of mitochondria. If the isolated mitochondrial samples carry cytoplasmic translated *β-galactosidase,* then it is likely to observe relatively the same level of translation for both the MT-C1 containing mitochondrial translation signal and MT-C1X lacking the mitochondrial translation signal. However, if the isolated mitochondrial samples carry mitochondrial translated *β-galactosidase,* then one may expect that the level of *β-galactosidase* for MT-C1X will be notably lower than that for MT-C1, which carries the mitochondrial translation signal. Illustrated in Figure 3a, it was observed that the *β-galactosidase* activity of isolated mitochondria from strains carrying mRNAs lacking mitochondrial translation signal, MT-C1X, was reduced by approximately 90% from that observed for the MT-C1construct that contains the mitochondrial translation signal. The *β-galactosidase* activity for MT-C1X was comparable to that for the control construct MT-C2 that lacks the *OXA1* 3′-UTR and is used to measure the background levels. As shown in Figure 3b, mRNA analysis using RT-qPCR revealed that, although MT-C1 and MT-C1X have relatively similar levels of *β-galactosidase* mRNA content, MT-C1X has a significantly lower translation level than MT-C1. This information points to the import of MT-C1X mRNA to the vicinity of the mitochondria, but with reduced translation levels, possibly because the construct lacks a mitochondrial translation initiation sequence, unlike MT-C1.

Next, we used antibiotics that specifically targeted different modes of translation (cytoplasmic vs. mitochondrial). As indicated, yeast mitochondrial translation resembles the prokaryotic mode of translation, whereas cytoplasmic translation represents that of eukaryotes. To further investigate if the *β-galactosidase* activity mediated by the MT-C1 construct is primarily influenced by mitochondrial translation, we utilized antibiotics thought to be specific to one mode of translation and not the other. To this end, we utilized cycloheximide and chloramphenicol. Cycloheximide is a eukaryotic translation inhibitor [26]. It interferes with translation elongation step by binding to the E-site of the eukaryotic ribosome. Chloramphenicol is a bacterial translation inhibitor. It disrupts the peptidyl transferase activity of ribosome during prokaryotic translation. We observed that, at high concentrations, (50 ng/mL and 1.5 mg/mL or higher for cycloheximide and chloramphenicol, respectively) both drugs reduced the *β-galactosidase* activity to near background levels for all constructs tested. At lower concentrations, however, the following observations were made. The presence of cycloheximide (45 ng/mL) in the media reduced the level of *β-galactosidase* activity mediated by MT-C2 and MT-C1X by approximately 83% and 95%, respectively (Figure 4a). This is in comparison with approximately 46% reduction for MT-C1. These observations suggest that translation mediated by MT-C1 construct seems to be more resistant to cycloheximide. In contrast, when the growth media was supplemented with 1.4 mg/mL of chloramphenicol, *β-galactosidase* produced by MT-C2 and MT-C1X was reduced by approximately 15% and 12%, indicating that they are not very sensitive to this concentration of chloramphenicol (Figure 4b). A reduction of 44% was observed for MT-C1, suggesting that translation mediated by MT-C1 construct appears to be more sensitive to chloramphenicol.

The discovery that all organisms have the same genetic codons gave rise to the notion that the genetic code is “universal”. However, there are some exceptions. The genome of the mitochondria contains most of these exceptions. The yeast mitochondrial codon exception is indicated in Appendix A. Exceptions to genetic codon implies certain variations in protein sequences if an mRNA is translated via cytoplasmic machinery versus mitochondrial translation. We analyzed the isolated mitochondrial samples on mass spectrometry to investigate any evidence for the mitochondrial translation of *β-galactosidase* mRNAs.

Table 1 shows the number of reads in each sample, both with mitochondrial protein ID (using mitochondrial codon exceptions) and cytoplasmic protein ID (using the universal codons). We observed that, in most of the reads processed by mitochondrial protein ID, MT-C1 that carries *OXA1* 3′-UTR along with the 5′ mitochondrial translation signal has the highest number of reads compared to MT-C2, which does not have the *OXA1* 3′-UTR but carries 5′-UTR from *COX2* mRNA. This supports the idea that *β-galactosidase* mRNAs are translated via mitochondrial machinery and that the presence of *OXA1* 3′-UTR is correlated to mitochondrial translation. For the cytosolic protein ID we recorded closer reads from all the samples. These may represent the background level of translation. We also performed mass spectrometry analysis on total protein extraction from yeast, but no mitochondrial reads were detected.

Concluding Remarks: The observations made in the current study provide genetic evidence for a correlation between the 3′-UTR of *OXA1* gene on a reporter mRNA and yeast mitochondrial translation. It seems that the reported ability of 3′-UTR of *OXA1* gene to direct mRNAs to the vicinity of mitochondrial may cause these mRNAs also to be translated by the mitochondrial translation machinery. There are two possibilities to explain how mitochondrial machinery can translate these mRNAs. They can either do so at the outer surface of the mitochondria, to where 3′-UTR of *OXA1* seem to direct mRNAs. Alternatively, it is possible that some of these mRNAs are directed into the mitochondria, where they can be translated. To our knowledge there is no report of mitochondrial translation outside of the mitochondria. It is also difficult to consider that large macromolecules, such as ribosomes, could be efficiently exported out of mitochondria. The export of translation machinery out of mitochondria is a more challenging task than mRNA import. Therefore, it seems likely that some of these mRNAs might be targeted into the mitochondria (Figure 5) and subjected to mitochondrial translation. It should also be noted that, although several steps have been taken to prevent cytoplasmic translation of these mRNAs, it remains a possibility that cytoplasmic translation might still play a role in explaining some of the observations made here. Further experiments are needed to study this possibility in detail.

Targeting RNAs into mitochondrial has been previously shown for 5S ribosomal RNA and microRNAs [12,27]. Similarly, a sequence from the *H1* RNA is reported to direct fusion of tRNA and mRNA molecules into mitochondria in different human cell lines [9]. The transport and translation of certain viral RNAs into the fungi mitochondria has also been reported [28,29,30]. It is predicted that the coding sequences of these viruses are non-translatable in the cytoplasm [31]. Details of how an mRNA could enter mitochondria are not clear. This might take place directly or indirectly via certain intermediator molecules, including protein channels (Figure 4). It is shown that RNA import can vary in different organisms but, overall, it relies on three important factors: (1). a selective signal or compartment within the RNA; (2). a mechanism to relocate RNA from its cytosolic location into the mitochondria surface; (3). a pathway to translocate RNA molecule into the mitochondria [27]. In plants, RNA import into mitochondria relies on unidentified cytosolic factors and requires Tom20 and Tom40 proteins for translocation into the outer membrane of mitochondria. An inner membrane translocation factor has not been discovered [27]. In yeast, tRNA import into mitochondria is facilitated by the translocase proteins of the outer and inner membranes, called TOM and TIM, respectively [27,32]. Little is known about the mechanism of RNA transport into human mitochondria. However, similar to what has been observed in yeast, RNA translocation in human cells seems to require membrane potential, possibly mediated by TOM or VDAC channel, across the mitochondrial membranes [32]. Another possible translocation mechanism is through intermembrane space by the activity of polynucleotide phosphorylase (PNPase) protein [33]. PNPase is shown to be able to recognize certain hairpin structures that seem to be import of RNA transport [33]. Structural studies permitted proposal of a mechanism by which human PNPase selects RNAs for mitochondrial import. Another translocation mechanism has been proposed in Leishmania, via a protein import pathway-independent mechanism called RNA Import Complex (RIC) [34]. It should be noted that the suggested model here is considered preliminary. Detailed biochemical assays should be employed to further study this proposed model. In future it would be of interest to further study the relationship between this 3′-UTR and mitochondria using other reporter genes and in other organisms including humans. The potential ability of *OXA1* 3′-UTR to direct an mRNA into the human mitochondria could have important implications for human health. A number of human genetic diseases are linked to defects in mitochondria encoded genes, or those nuclear genes whose products are directed into mitochondria [35,36]. In this context, the ability to direct a corrective mRNA into mitochondria could provide an appealing therapeutic opportunity.

## Figures and Tables

**Figure 1 jof-09-00445-f001:**
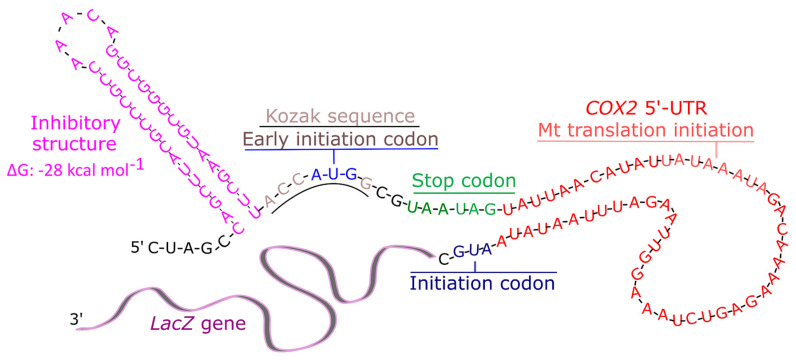
Schematic representation of the *β-galactosidase* mRNA designed to limit cytoplasmic translation. Various features of the 5′-UTR for this mRNA are shown in different colors. Elements designed to limit cytoplasmic translation include an inhibitory structure (pink), an early initiation codon (light blue) within Kozak sequence (light gray) and early tandem stop codons (green). *COX2* 5′-UTR (red) mediates mitochondrial translation; its key translation initiation sequence (Mt translation initiation) is indicated (light red).

**Figure 2 jof-09-00445-f002:**
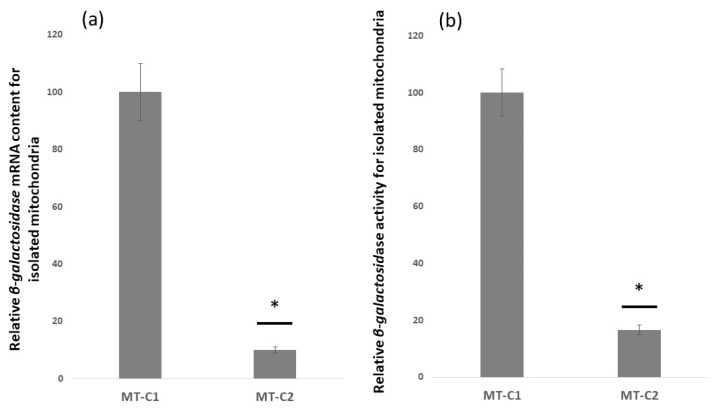
mRNA content and *β-galactosidase* expression analysis of isolated mitochondria from different yeast strains. (**a**) RT-PCR analysis indicated that in the absence of *OXA1* 3′-UTR (MT-C2) *β-galactosidase* mRNA content is significantly reduced in isolated mitochondria. The values are normalized to that of the MT-C1 construct that carries *OXA1* 3′-UTR. *PGK1* mRNA was used as a control and all values are related to that. (**b**) *β-galactosidase* activity of isolated mitochondria is highly reduced in the absence of *OXA1* 3′ UTR (MT-C2). Values are normalized to that for MT-C1 set at 100. Each experiment is repeated at least three times. Error bars represent standard deviation. * represents statistically significant results compared to the MT-C1. *t*-test analysis (*p*-value ≤ 0.05) was used to compare differences.

**Figure 3 jof-09-00445-f003:**
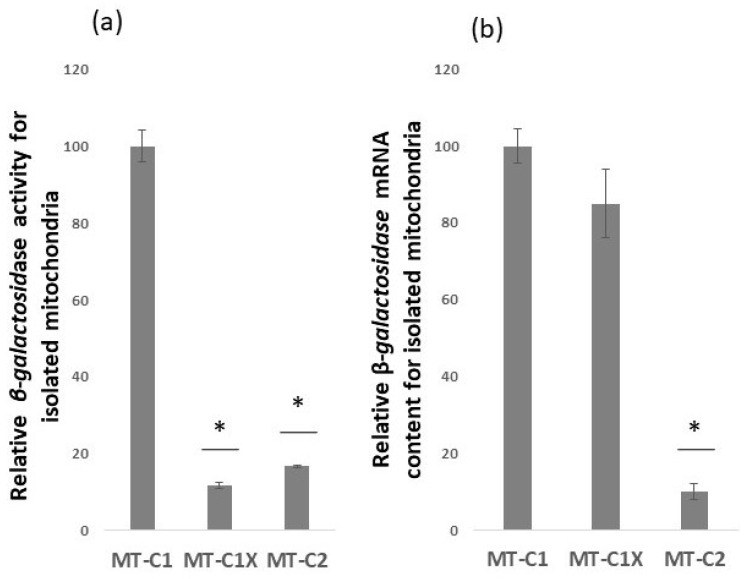
Analysis of relative *β-galactosidase* activities and relative mRNA content of isolated mitochondria. (**a**) *β-galactosidase* activity of isolated mitochondria is highly reduced in the absence of *COX2* mitochondrial translation initiation signal MT-C1X in comparison to the MT-C1 construct that contains the translation initiation signal. (**b**) To relate *β-galactosidase* activity to translation level, mRNA content of the constructs was compared via RT-qPCR analysis. The result showed similar levels of *β-galactosidase* mRNA for MT-C1 and MT-C1X, whereas the MT-C2 mRNA level was significantly lower, possibly in the absence of the OXA1 3′-UTR sequence. Error bars represent standard deviation. * Represents statistically significant results compared to the MT-C1. *t*-test analysis (*p*-value ≤ 0.05) was used to compare differences.

**Figure 4 jof-09-00445-f004:**
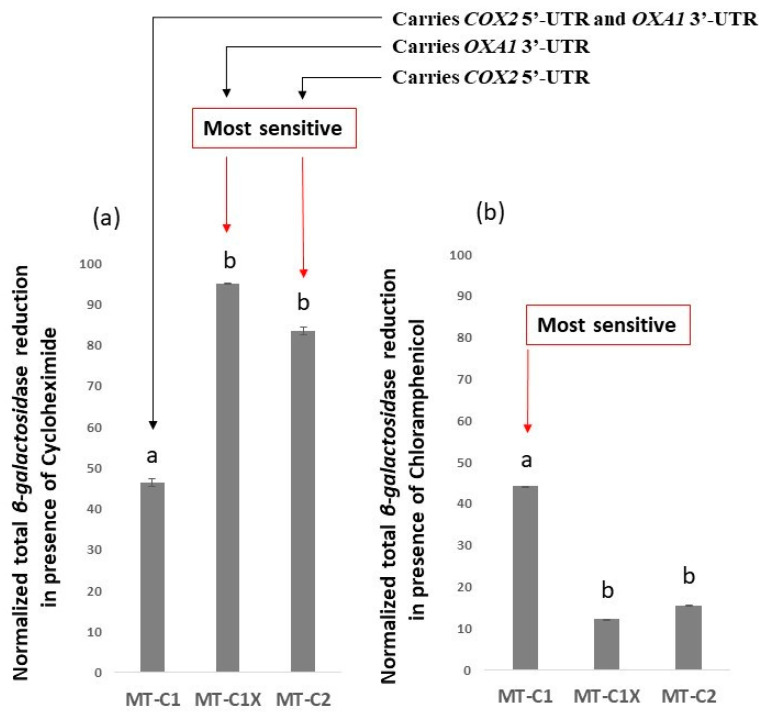
(**a**) In the presence of 45 ng/mL of cycloheximide, a cytoplasmic translation inhibitor, *β-galactosidase*, mediated by MT-C1X and MT-C2 showed the most sensitivity. (**b**) In contrast, when 1.4 mg/mL of chloramphenicol, a prokaryotic translation inhibitor, was added, *β-galactosidase* activity mediated by MT-C1 showed the most sensitivity. The values in (**a**) are related to the value of MT-C1 and values in (**a**,**b**) are normalized to the translation of the same construct in the absence of the corresponding translation inhibitory compound. Each experiment is repeated at least 3 times. Error bars represent standard deviation. a and b, represent statistically different values. One-way ANOVA (*p*-value ≤ 0.05) was used to compare the differences.

**Figure 5 jof-09-00445-f005:**
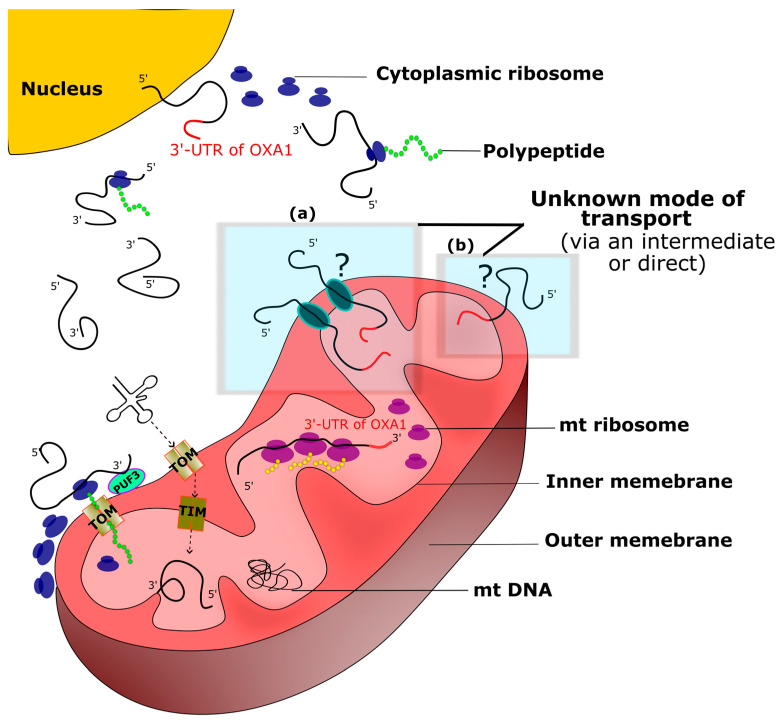
A model for the role of 3′-UTR of *OXA1* gene. Different modes of RNA entry into mitochondria are suggested by (a) and (b), where (a) is through membrane proteins and (b) demonstrates a direct method. In addition to its reported activity in recruiting mRNAs to the vicinity of mitochondria, 3′-UTR of *OXA1* gene may also direct mRNAs into mitochondria via a currently unknown mechanism. The majority of mitochondrial proteins are of nuclear origin and are imported into mitochondria after translation. The 3′-UTR of yeast *OXA1* mRNA can direct mRNAs to the vicinity of mitochondria via an interaction with mitochondria outer surface protein Puf3. The newly translated polypeptides, near the vicinity of mitochondria, are directed into mitochondria using general translocase of the outer membrane (TOM) complex. TOM and TIM complex are also shown to mediate the translocation of tRNAs. Here, we propose that the 3′-UTR of *OXA1* mRNA might also mediate the entry of mRNAs into mitochondria, where they are translated by mitochondrial translation machinery.

**Table 1 jof-09-00445-t001:** Total peptide count reads of MT-C1, MT-C2, and MT-C1X mitochondrial proteins using Mass spectrometry. Runs were performed with mitochondrial specific protein IDs and cytoplasmic specific protein IDs. Peptide sequence for each are shown below. We were able to detect mitochondrial translation in the samples. As expected, MT-C1 that carries *OXA1* 3′-UTR and the 5′ mitochondrial translation signal shows the highest reads compared to MT-C2 (carrying 5′ mitochondrial translation signal only) and MT-C1X (carrying *OXA1* 3′-UTR and a mutated 5′ mitochondrial translation signal). There are some reads in which MT-C1 is not showing the highest read, probably due to co-purification and background translation levels. For the cytosolic protein ID, we observe a closer range of reads for the three samples.

	Total Peptide Count		
	MT-C1	MT-C1X	MT-C2
Mitochondria Protein ID			
TTFMSRK	11	---	---
DQFTRAPTDNDMGVSEATR	43	---	49
QTTTPTR	45	75	67
RWQFNR	154	---	53
HQQQFFQFRTSGQTMEVTSEYTFR	236	33	31
TPHPATTEAK	21	---	---
VDEDQPFPAVPKWSMK	37	---	13
VTRMVQR	53	58	75
WTPAMSERVTR	86	11	55
TWSAEIPNTYR	319	---	191
FNDDFSRAVTEAEVQMCGETRDYTR	11	---	3
TAVMVTR	17	36	63
MTMMTDSTAVVTQRR	52	---	23
Cytoplasmic protein ID			
GDFQFNISR	---	41	---
QSGFLSQMWIGDKK	---	34	---
QLLTPLR	44	46	97
MVQRDR	45	75	67
AGENRLAVMVLR	30	40	37
LAVMVLRWSDGSYLEDQDMWRMSGIFR	34	3	27

## Data Availability

No new data were created.

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
