# Peer review of "A Correlation between 3′-UTR of OXA1 Gene and Yeast Mitochondrial Translation"

_jof, 2023, doi:10.3390/jof9040445_

Round 1

Reviewer 1 Report

The aim of the peer-reviewed manuscript is to shed more light on whether the 3'-UTR in eukaryotic mRNA is associated with its translocation into mitochondria and whether such mRNA can be translated by the mitochondrial translation machinery. To answer this difficult question the authors designed original mRNAs constructs containing a reporter β-galactosidase coding sequence supplemented with 3'-UTR of OXA1 mRNA (downstream, which is believed to be responsible for bringing the nuclear mRNA closer to the mitochondria) and upstream regulatory elements for initiation or suppression of mitochondrial or eukaryotic translation. Using yeast as a model for studying translation, the authors obtained clear cut evidence of correlation between 3'-UTR of OXA1 and the translation of β-galactosidase mRNA in mitochondria.

The study was conducted with modern methodology and research technique. The experiments are described in great detail, which makes it possible to be reproduced without any need of additional methodological literature. The results are convincing and well illustrated with tables, figures and comprehensive legends. They fully support the authors' working hypothesis, fill some white spots in our knowledge of the interrelationship of cytoplasmic-mitochondrial translation, and are of interest to a wide range of readers working in the fields of molecular biology, molecular genetics, cell biology and mitochondrial diseases.

Since all experiments were performed with S. cerevisiae (strain BY4741), the manuscript is eligible for the journal Yeast Genetics. English language is appropriate and understandable?

Author Response

We would like to start by thanking the anonymous reviewers for generously dedicating their time and offering insightful comments to enhance the quality of this manuscript.

Reviewer 2 Report

Summary:

This study investigates the import and translation of synthetic RNAs into yeast mitochondria. The authors created a synthetic mRNA containing a mitochondrial mRNA targeting domain (Oxa1 3’UTR), a mitochondrial translation initiation sequence (Cox2 5’UTR) followed by a reporter gene (lacZ) and sequences that should interfere with cytosolic translation. The authors show that this synthetic mRNA can be isolated with mitochondria in ways that are dependent on the Oxa1 3’UTR. The authors investigate translation from synthetic mRNAs by following B-gal activity in isolated mitochondria and in cells expressing cytosolic and mitochondrial translation inhibitors, and by analyzing peptide content from mitochondrial factions using mass-spec. The authors conclude that at least some translation of the synthetic mRNA is occurring by mitochondrial ribosomes.

General comments:

The ability to target mRNAs into the mitochondria could offer important therapeutic strategies for treating mitochondrial disorders. The synthetic RNA created by the authors is clever, and the authors convincingly show that the Oxa1 3’-UTR helps to localize the synthetic mRNA to the mitochondria. It’s not clear why the authors think that this mRNA should be imported into the mitochondria, and there are concerns about the conclusion that the synthetic mRNA is translated by mitochondrial ribosomes. The following comments and suggested experiments are aimed at increasing the value of this study.

Comments/concerns:

1.     Could the authors expand on why they think this mRNA (or any mRNA containing the Oxa1 3’UTR) should be imported into the mitochondria? Do you expect that any nuclear-encoded mRNA that is localized to the OM of the mitochondria to also be imported into the mitochondria, or is there something special about the Oxa1 mRNA?  

2.     In Fig 3A, the authors argue that the reduction of B-gal activity in mitos isolated from cells expressing MT-C1X as compared to C1 is due to a reduction of (mitochondrial) translation. The loss of B-gal activity could be due to a reduction in mRNA stability. To relate B-gal activity to translation levels, the mRNA levels of C1 vs C1X should be compared (similar to Fig 2A).

3.     Currently, it’s difficult to assess how meaningful the comparisons of normalized B-gal reduction in treated and untreated cells are. How do the whole cell B-gal activities compare to the B-gal activities in purified mitochondria? If the whole cell B-gal activities in cells expressing C1X and C2 constructs are very low (and it’s not clear to me what they would be), than a large reduction in normalized activity may in fact be a very small change in B-gal activity. How do they compare to background B-gal activity levels in treated and untreated cells expressing an empty plasmid (eg. pRS416)? Statistical comparisons of all treated and untreated conditions should be included.

4.     Cycloheximide and chloramphenicol had similar normalized total B-gal reductions in C1 (activity was reduced to ~55% of total levels). Are these effects additive? Does adding cycloheximide and chloramphenicol simultaneously eliminate B-gal activity? If so, it would help to argue that translation is occurring by both cytosolic and mitochondrial ribosomes.

5.     I do not have experience with mass-spec analyses but I am concerned that mitochondrially-translated lacZ peptides were observed in all samples. The higher numbers of certain mitochondrially-translated peptide counts in C1X and C2 than in C1 were explained as “probably due to co-purification and background translation levels”. The authors also conclude that low B-gal levels in C1X and C2 purified mitos is due to low translation. Could the authors elaborate on this? Is it possible to compare these lacZ peptides to other mitochondrially translated peptides to show that there is an enrichment in lacZ peptides in C1 samples?

6.     Additional experiments would help to support the idea that translation of this synthetic mRNA is translated by mitochondrial ribosomes. Here are 2 suggestions:

a. The authors could alter one or more of the tryptophan codons in the lacZ gene (there are at least 3 near the N-terminus) to match the mitochondrial codon usage (UGA=stop by cytosolic ribosomes and Trp by mitochondrial ribosomes).  Any B-gal activity by such constructs would necessarily be due to mitochondrial translation.

b. It’s unlikely that without targeting sequences, B-gal synthesized in the cytosol would be imported into the mitochondria or that mitochondrially-translated B-gal would be exported out of the mitochondria. Therefore, B-gal activity in isolated mitochondria containing mitochondrially-translated B-gal should be resistant to proteinase K (and could be included as controls to the experiments shown in 2B or 3A).

minor editorial comments:

1. Line 57 lacks a reference.

2. line 43 Patients should be lowercase

Author Response

Summary:

This study investigates the import and translation of synthetic RNAs into yeast mitochondria. The authors created a synthetic mRNA containing a mitochondrial mRNA targeting domain (Oxa1 3’UTR), a mitochondrial translation initiation sequence (Cox2 5’UTR) followed by a reporter gene (lacZ) and sequences that should interfere with cytosolic translation. The authors show that this synthetic mRNA can be isolated with mitochondria in ways that are dependent on the Oxa1 3’UTR. The authors investigate translation from synthetic mRNAs by following B-gal activity in isolated mitochondria and in cells expressing cytosolic and mitochondrial translation inhibitors, and by analyzing peptide content from mitochondrial factions using mass-spec. The authors conclude that at least some translation of the synthetic mRNA is occurring by mitochondrial ribosomes.

General comments:

The ability to target mRNAs into the mitochondria could offer important therapeutic strategies for treating mitochondrial disorders. The synthetic RNA created by the authors is clever, and the authors convincingly show that the Oxa1 3’-UTR helps to localize the synthetic mRNA to the mitochondria. It’s not clear why the authors think that this mRNA should be imported into the mitochondria, and there are concerns about the conclusion that the synthetic mRNA is translated by mitochondrial ribosomes. The following comments and suggested experiments are aimed at increasing the value of this study.

Our Response: Thank you for your comment. The main conclusion that we have from our observation is that OXA1 mRNA 3’-UTR can direct a reporter mRNA to the vicinity of the mitochondria (in confirmation of other studies) and that the translation of this mRNA has signatures of mitochondrial translation. Here we report a correlation between OXA1 mRNA 3’-UTR on an mRNA and mitochondrial translation. One possible explanation is that mitochondrial translation may take place in the vicinity of mitochondria and that mitochondrial translation machinery may get exported out of the mitochondria to do this. A second possibility is that OXA1 mRNA 3’-UTR containing reporter mRNA may get translated inside mitochondria and that the mRNA is imported into mitochondria. mRNA is significantly smaller and simpler to cross mitochondrial membrane and there is some evidence in literature so mRNA import into mitochondria. However in the current study, we do not have enough evidence for either mechanism and more investigation is needed. We feel this is out of the scope of the current study.

The majority of the comments below are directed towards this point. We hope we have clarified our intention and that mRNA import is only a possible way to explain these observations.

Comments/concerns:

  1. Could the authors expand on why they think this mRNA (or any mRNA containing the Oxa1 3’UTR) should be imported into the mitochondria? Do you expect that any nuclear-encoded mRNA that is localized to the OM of the mitochondria to also be imported into the mitochondria, or is there something special about the Oxa1 mRNA?  

Our response: Please see our response to the General Comment by Rev 2 above. In addition Oxa1p is a conserved membrane protein that facilitates the insertion of mitochondrion precursor into the inner membrane of the mitochondria (1). OXA1 3’-UTR has been previously shown to have a crucial role in localizing mRNAs into the vicinity of mitochondria (2–4). In the current study, we used this 3’-UTR sequence of OXA1, to direct our construct towards mitochondria membrane. Several nuclear-encoded mRNAs are located in the vicinity of mitochondria, translated, and then imported into mitochondria via membrane proteins, according to what is currently known. Numerous small mRNAs and RNAs have also been demonstrated to be imported into mitochondria, though the precise process is unknown. Answering the question of weather mitochondrial translation can take place in the OM of mitochondria is a very interesting question. Unfortunately it is out of the scope of the current study and we do not have enough evidence to comment on it either way. Having said that the export of translation machinery out of mitochondrial appears to be a more challenging task than mRNA import. We modified our conclusion to emphasize this point (366-367).

  1. In Fig 3A, the authors argue that the reduction of B-gal activity in mitos isolated from cells expressing MT-C1X as compared to C1 is due to a reduction of (mitochondrial) translation. The loss of B-gal activity could be due to a reduction in mRNA stability. To relate B-gal activity to translation levels, the mRNA levels of C1 vs C1X should be compared (similar to Fig 2A).

Our response: This is valid. We have now added the qPCR data to the new Fig 3b.

  1. Currently, it’s difficult to assess how meaningful the comparisons of normalized B-gal reduction in treated and untreated cells are. How do the whole cell B-gal activities compare to the B-gal activities in purified mitochondria? If the whole cell B-gal activities in cells expressing C1X and C2 constructs are very low (and it’s not clear to me what they would be), than a large reduction in normalized activity may in fact be a very small change in B-gal activity. How do they compare to background B-gal activity levels in treated and untreated cells expressing an empty plasmid (eg. pRS416)? Statistical comparisons of all treated and untreated conditions should be included.

Our response: This is a valid point. Isolated mitochondria is highly enriched and should not be compared to the whole cell analysis. Isolated mitochondrial was used to reduce the influence of cytoplasmic produced Bgal in the expression analysis and it should not be mixed with drug treatment analysis. For clarity, we divided the old Fig 3, into a new Fig 3 and a new Fig 4. The new Fig 3 shows the Bgal activity and mRNA content of the constructs in isolated mitochondria and the new Fig 4 shows the whole cell activity of Bgal in different construct under influence of chloramphenicol and cycloheximide.

  1. Cycloheximide and chloramphenicol had similar normalized total B-gal reductions in C1 (activity was reduced to ~55% of total levels). Are these effects additive? Does adding cycloheximide and chloramphenicol simultaneously eliminate B-gal activity? If so, it would help to argue that translation is occurring by both cytosolic and mitochondrial ribosomes.

Our response: This is an interesting comment, thank you. We have tried this already. Unfortunately adding both drugs drastically affected cell survival. Lower concentrations of individual drugs had little effects and higher concentrations drastically lowered viability.

  1. I do not have experience with mass-spec analyses but I am concerned that mitochondrially-translated lacZ peptides were observed in all samples. The higher numbers of certain mitochondrially-translated peptide counts in C1X and C2 than in C1 were explained as “probably due to co-purification and background translation levels”. The authors also conclude that low B-gal levels in C1X and C2 purified mitos is due to low translation. Could the authors elaborate on this? Is it possible to compare these lacZ peptides to other mitochondrially translated peptides to show that there is an enrichment in lacZ peptides in C1 samples?

Our response: Due to technical limitation of MS, the number of peptide reads plays a key role for the validity of the observations. In these samples, Bgal is overexpressed. A representative control would be the number of peptide reads, which are specific to cytoplasmic translation. This is provided at the bottom of the Table and serves as a control. 

  1. Additional experiments would help to support the idea that translation of this synthetic mRNA is translated by mitochondrial ribosomes. Here are 2 suggestions:
  2. The authors could alter one or more of the tryptophan codons in the lacZ gene (there are at least 3 near the N-terminus) to match the mitochondrial codon usage (UGA=stop by cytosolic ribosomes and Trp by mitochondrial ribosomes).  Any B-gal activity by such constructs would necessarily be due to mitochondrial translation.
  3. It’s unlikely that without targeting sequences, B-gal synthesized in the cytosol would be imported into the mitochondria or that mitochondrially-translated B-gal would be exported out of the mitochondria. Therefore, B-gal activity in isolated mitochondria containing mitochondrially-translated B-gal should be resistant to proteinase K (and could be included as controls to the experiments shown in 2B or 3A).

Our response:  Thank you for these suggestions. These are very good experiments for a future study. We would certainly consider them.

minor editorial comments:

  1. Line 57 lacks a reference. Reference is added.

  1. line 43 Patients should be lowercase. It is corrected.

  1. Krüger V, Deckers M, Hildenbeutel M, Van Der Laan M, Hellmers M, Dreker C, Preuss M, Herrmann JM, Rehling P, Wagner R, et al. The mitochondrial oxidase assembly protein1 (Oxa1) insertase forms a membrane pore in lipid bilayers. J Biol Chem. 2012;287(40):33314–26.
  2. Gadir N, Haim-vilmovsky L, Kraut-cohen J, Gerst JE. Localization of mRNAs coding for mitochondrial proteins in the yeast Saccharomyces cerevisiae. RNA Biol [Internet]. 2011;1551–65. Available from: https://rnajournal.cshlp.org/content/17/8/1551.long
  3. Sylvestre J, Margeot A, Jacq C, Dujardin G. The Role of the 3’ Untranslated Region in mRNA Sorting to the Vicinity of Mitochondria Is Conserved from Yeast to Human Cells. Mol Biol Cell [Internet]. 2003;14(September):3848–56. Available from: ncbi.nlm.nih.gov/pubmed/12972568
  4. Fox TD. Mitochondrial protein synthesis, import, and assembly. Yeastbook, cell Struct Traffick [Internet]. 2012;192(December):1203–34. Available from: https://doi.org/10.1534/genetics.112.141267

Round 2

Reviewer 2 Report

Thank you for addressing the concerns described in the previous review. The addition of the new Fig 3b is nice.

1. Could you describe the statistical tests used in Fig 4? (Presumably post-hoc Tuckey hsd )

2. The argument seems to be that translation of the synthetic RNA has to be due to mitochondrial translation, either within or outside the mitochondria. Isn't it possible that the synthetic RNA is translated by cytosolic ribosomes in the vicinity of the mitochondria?  I see that B-gal activity is more sensitive to chloramphenicol than cycloheximide at low concentrations but think that this chloramphenicol sensitivity could be indirect, perhaps by altering the effectiveness of mitochondrial membrane-associated cytosolic translation. High levels of both antibiotics blocked Bgal activity, indicating that there is some ambiguity with this assay. In the absence of experiments that prove mitochondrial translation, the authors should at least provide cytosolic translation (even if you believe it is unlikely) as a possibility. 

Author Response

Response to Reviewers’ Comments

We would like to start by thanking the anonymous reviewers for generously dedicating their time and offering insightful comments to enhance the quality of this manuscript.

Thank you for addressing the concerns described in the previous review. The addition of the new Fig 3b is nice.

1.Could you describe the statistical tests used in Fig 4? (Presumably post-hoc Tuckey hsd )

Our Response: Yes, the statistical test is added to the Fig 4 description.

  1. The argument seems to be that translation of the synthetic RNA has to be due to mitochondrial translation, either within or outside the mitochondria. Isn't it possible that the synthetic RNA is translated by cytosolic ribosomes in the vicinity of the mitochondria?  I see that B-gal activity is more sensitive to chloramphenicol than cycloheximide at low concentrations but think that this chloramphenicol sensitivity could be indirect, perhaps by altering the effectiveness of mitochondrial membrane-associated cytosolic translation. High levels of both antibiotics blocked Bgal activity, indicating that there is some ambiguity with this assay. In the absence of experiments that prove mitochondrial translation, the authors should at least provide cytosolic translation (even if you believe it is unlikely) as a possibility. 

Our Response: This possibility is now added to the manuscript.  
